# Ageing, functioning patterns and their environmental determinants in the spinal cord injury (SCI) population: A comparative analysis across eleven European countries implementing the International Spinal Cord Injury Community Survey

Carla Sabariego[1,2,3]*, Cristina Ehrmann[1], Jerome Bickenbach[1,2,3], Diana Pacheco Barzallo[1,2,3], Annelie Schedin Leiulfsrud[4,5], Vegard Strøm[6], Rutger Osterthun[7,8], Piotr Tederko[9], Vanessa Seijas[1,2,3], Inge Eriks-Hoogland[1,2,10], Marc Le Fort[11], Miguel A. Gonzalez Viejo[12], Andrea Bökel[13], Daiana Popa[14], Yannis Dionyssiotis[15], Alessio Baricich[16], Alvydas Juocevicius[17], Paolo Amico[18], Gerold Stucki[1,2,3]

1 Swiss Paraplegic Research, Nottwil, Switzerland, 2 Faculty of Health Sciences and Medicine, University of Lucerne, Lucerne, Switzerland, 3 Faculty of Health Sciences and Medicine, Center for Rehabilitation in Global Health Systems, University of Lucerne, Lucerne, Switzerland, 4 Department of Spinal Cord Injuries, St Olav University Hospital, Trondheim, Norway, 5 Department of Neuromedicine and Movement Science, Norwegian University of Science and Technology, Trondheim, Norway, 6 Department of Research, Sunnaas Rehabilitation Hospital, Nesodden, Norway, 7 Rijndam Rehabilitation, Rotterdam, the Netherlands, 8 Erasmus MC, University Medical Center Rotterdam, Rotterdam, the Netherlands, 9 Department of Rehabilitation, Medical University of Warsaw, Warsaw, Poland, 10 Swiss Paraplegic Centre, Nottwil, Switzerland, 11 CHU Nantes—Hôpital Saint-Jacques, Nantes, France, 12 Hospital Vall d'Hebron Barcelona, Barcelona, Spain, 13 Department of Rehabilitation Medicine, Hannover Medical School, Hanover, Germany, 14 Rehabilitation Hospital Felix-Spa, Bihor, Romania, 15 Spinal Cord Injury Rehabilitation Clinic, University of Patras, Rio Patras, Greece, 16 Department of Health Sciences, Università del Piemonte Orientale, Novara, Italy, 17 The Center of Rehabilitation, Physical and Sports Medicine, Vilnius University Hospital Santaros Klinikos, Vilnius, Lithuania, 18 Policlinico Hospital of Bari, Bari, Italy

* carla.sabariego@unilu.ch

## Abstract

### Background

As the European population with Spinal Cord Injury (SCI) is expected to become older, a better understanding of ageing with SCI using functioning, the health indicator used to model healthy ageing trajectories, is needed. We aimed to describe patterns of functioning in SCI by chronological age, age at injury and time since injury across eleven European countries using a common functioning metric, and to identify country-specific environmental determinants of functioning.

### Methods

Data from 6'635 participants of the International Spinal Cord Injury Community Survey was used. The hierarchical version of Generalized Partial Credit Model, casted in a Bayesian framework, was used to create a common functioning metric and overall scores. For

**Data availability statement:** The datasets presented in this article are not publicly available because the countries participating in the InSCI survey hold the rights for data use. Requests for data use of third parties need to be negotiated with respective countries. Requests to access the datasets should be directed to insci@paraplegie.ch.

**Funding:** The author(s) received no specific funding for this work.

**Competing interests:** The authors have declared that no competing interests exist.

each country, linear regression was used to investigate associations between functioning, chronological age, age at SCI or time since injury for persons with para- and tetraplegia. Multiple linear regression and the proportional marginal variance decomposition technique were used to identify environmental determinants.

## Results

In countries with representative samples older chronological age was consistently associated with a decline in functioning for paraplegia but not for tetraplegia. Age at injury and functioning level were associated, but patterns differed across countries. An association between time since injury and functioning was not observed in most countries, neither for paraplegia nor for tetraplegia. Problems with the accessibility of homes of friends and relatives, access to public places and long-distance transportation were consistently key determinants of functioning.

## Conclusions

Functioning is a key health indicator and the fundament of ageing research. Enhancing methods traditionally used to develop metrics with Bayesian approach, we were able to create a common metric of functioning with cardinal properties and to estimate overall scores comparable across countries. Focusing on functioning, our study complements epidemiological evidence on SCI-specific mortality and morbidity in Europe and identify initial targets for evidence-informed policy-making.

## Introduction

Spinal Cord Injury (SCI) refers to a traumatic or non-traumatic damage of neural elements within the spinal canal and is characterized by the loss or impairment of motor, sensory, and or autonomic functions below the level of the of the injury [1]. Traumatic SCI is caused by accidents, for instance, traffic or work-related accidents, violence, and falls while non-traumatic SCI is associated with degenerative, inflammatory, neoplastic and infectious health conditions. [2]. Data from the Global Burden of Disease (GBD) shows that the 2016 age-standardized prevalence of SCI rates per 100,000 persons (95% uncertainty interval) were among the highest for Western Europe with 854 (780–945) and high in Central Europe with 597 (549–653) [3,4]. Survival, health status, and quality of life of persons with SCI very much depend on the context where the person lives [5]. The context is shaped by factors such as accessibility of the place of living, access to appropriate acute and post-acute medical care, availability of supportive policies, and negative attitudes of the general population regarding SCI, among others. While in high-income countries (HICs), in which the context is more frequently facilitating, persons with SCI can age and thrive, in low- and middle-income countries (LMICs) a SCI might lead to a short and limited life given a myriad of environmental barriers.

In Europe, SCI among ageing population is expected to increase in absolute numbers but also to change its incidence and prevalence patterns. The share of older people in Europe is projected to grow to 28.5% in 2050 [3] and by far the largest cause of SCI incidence in Western and Central Europe are falls–a common phenomenon in the elderly population [3]. Indeed, while young adult injuries are frequently caused by high-intensity trauma, SCI in adults and the elderly are mainly caused by low-intensity trauma like falling from heights of less than one meter, or by non-traumatic SCI associated with health conditions [6,7]. Large American

studies evaluating demographic changes in SCI show that mean age at injury associated with falls and accidents has been steadily increasing in the past decades [8], from ca. 28 years in the 1970s to ca. 42 years between 2010 and 2014 [9]. This trend is also observed in regional and national European studies, for instance in Norway [10], the Netherlands [11] and Finland [12] for traumatic SCI, in Switzerland [13] for non-traumatic SCI and for both in Scotland [14]. The increase of older populations at risk of SCI, the continuous shift in age at injury as well as longer life expectancy due to medical developments and better access to health and supportive services will all pose considerable challenges to health and social European systems.

An adequate response to the needs of persons ageing with SCI require comprehensive research on the topic [15]. Ageing with SCI is complex because this group besides age-related conditions, have already a multimorbid situation, both of which accelerate the ageing process [16]. Currently, we observe that the mean age of the lesion is increasing due to an increment in the number of falls and domestic accidents. At the same time, the increasing life expectancy of persons with SCI rises the propensity of this group of facing more health complications [17]. For these reasons, related literature emphasizes the importance to understand how the multiple complications observed on persons with SCI are related to their chronological age, to the age at the onset of the injury, and to the time with the injury [17,18].

After an injury, many body functions are affected and changes are multifold, including declines in muscle strength, bone density and metabolic rate [17]. A range of secondary conditions, including cardiovascular disease, respiratory and urinary tract infections, neuropathic pain and pressure ulcers are also frequent [19]. Additionally, persons with SCI, compared to the general population, are more prone to suffer from infections and fractures [20,21]. In this context, age at the time of the injury matters because people who acquire SCI at a younger age are likely to have a better health status than people that acquired the injury at an older age, who are more resilient and, in general, experience a faster and greater recovery [18]. In contrast, persons who acquire SCI at an older age have more nontraumatic injuries, cervical injuries, complications, and higher mortality risk [22]. Duration of injury is relevant because of the negative effect of length of exposition time to bladder and bowel problems, fracture risk, pressure sores, recurrent infections, high medication use, and immune system changes, among others. The continuous experience of complications and their cumulative effect over time can augment health decrements associated with the usual ageing process. Moreover, there is a negative association between duration of injury and time lived with mobility limitations, including for instance lack of or insufficient physical activity in older wheelchair users [23]. The physical, social and political context of the person is likely to considerably affect the ageing process as well [18]. Examples include surroundings that are not wheelchair friendly, restrict mobility and worsening sarcopenia, higher levels of social isolation (important risk factor for dementia in older adults) in non-barrier free neighborhoods as well as policies that do not ensure access to the specialized health care needed by persons with SCI.

Functioning, as defined in the World Health Organization (WHO) International Classification of Functioning, Disability and Health (ICF), is a key health indicator for comprehensive ageing research [24]. Functioning encompasses the dimensions of the lived experience of health, from body functions, such as mental and cardiovascular functions, to activities, such as moving around and self-care, and participation domains, such as work and community involvement [24]. Additionally, functioning is understood as the outcome of the interaction between health conditions, personal-psychological factors [25] and a range of features of the person's context, such as the accessibility of the place of living, family support, social attitudes, and access to health care [24]. For a complex health condition like SCI it is essential to have information about this full lived experience, contextualized to the details of the person's situation. Moreover, the notion of functioning is ideally suited for ageing research as it accounts

for the synergistic effect of multimorbidity. Functioning has been frequently used in ageing research to model and examine trends in morbidity [26] and more recently to model so-called healthy ageing trajectories [27]. The use of functioning in SCI research is common, and is found in visual representations of associations using graphical modeling [28], modelling of functioning trajectories during first rehabilitation [29] and analyses of the impact of environmental barriers [30–32].

Functioning as an overall indicator to model healthy ageing trajectories over time, however, has not yet been used to obtain a broader understanding of ageing with SCI. The functioning indicator is built by condensing its components of body functions and structures, activities and participation domains, as defined in the ICF, into a single indicator and metric. This allows the study of the impact of its determinants, i.e. personal-psychological and environmental factors, on the level of functioning of specific populations. The functioning indicator has therefore the potential to contribute to the understanding of the complexity of ageing with SCI, enabling the identification of its most relevant personal-psychological and environmental predictors. Using data collected in the International Spinal Cord Injury Community Survey (InSCI) [33] in eleven European countries, the objectives of this study are, therefore, 1) to describe patterns of functioning in SCI by chronological age, age at injury and time since injury across countries using a common functioning metric and an overall score of functioning, and 2) to identify country-specific environmental determinants of overall functioning that can serve evidence-informed policy making.

## Materials and methods

Defining who the ageing population is, in terms of chronological age, is a challenge. While the European Commission defines in the Ageing Europe report [34] old persons as 65+ and very old persons as 85+, most ageing cohort studies recruit 50+ participants. The value of such cut-offs has been challenged as the health state associated with a certain age can vary considerably depending across countries [35]. Due to the interference of SCI on the "normal" ageing process, defining cut-offs for SCI is even more challenging. We therefore decided not to pre-define any chronological age cut-off and to carry out analyses and present results for the complete InSCI sample by chronological age, age at the time of the injury and duration of the injury.

### Ethics statement

This study uses data from the first InSCI community survey collected in 11 countries. In the context of data collection, compliance with national laws and regulatory approvals by Institutional Review Boards or Ethical Committees was mandatory for all countries and conform to the Helsinki Declaration. Each national study group was responsible to ensure this compliance. Informed consent was obtained from all participants and/or their legal guardian in accordance with national regulations. All methods were performed in accordance with the Declaration of Helsinki. The full name of the Institutional Review Boards or Ethical Committees who approved the study are: Comité de Protection des Personnes (France), Ethic Committee of Hannover Medical School (Germany), Scientific/Ethical Committee of General Hospital 'G. Gennimatas' Athens (Greece), Comitato Etico Interaziendale AOU 'Maggiore della Carità' di Novara, ASL BI, ASL NO, ASL VCO (Italy), Vilnius Regional Committee for the Ethics of Biomedical Research (Lithuania), Medical Ethics Board University Medical Center Utrecht (Netherlands), Regional Committee for Medical and Health Research Ethics, South East (Norway), Bioethical Committee of the Medical University of Lodz (Poland), Ethical Committee of Rehabilitation Hospital Felix Spa (Romania), Ethical Committee of Hospital

Universitari Vall d'Hebron, Hospital Universitario de Cruces, Hospital Universitario Materno Infantile de Gran Canaria, Hospital Universitario Virgen del Rocio (Spain) and Ethical Committee of Northern and Central Switzerland (Switzerland). Reference numbers of all ethical approvals are included in S1 Table.

## Study design

InSCI has a cross-sectional design, uses the ICF as a reference framework and collects functioning data of persons with SCI in 22 countries worldwide [36]. The InSCI data model includes ICF categories listed in the brief ICF core set for SCI in the long-term context [37] and in the ICF Core Set for rehabilitation [38]. For the analysis, we used data collected in Norway, the Netherlands, France, Germany, Greece, Italy, Spain, Poland, Romania, Lithuania and Switzerland. As many countries lacked registries of persons with SCI, we only count with random samples from Norway, Germany, the Netherlands, Poland and Switzerland. For the remaining countries, convenient sampling was the only possibility to recruit participants, either through contact in health care facilities, government agencies and pre-existing databases from patient associations. To invite persons with SCI to participate in the survey, the national research teams used invitation letters, e-mails, phone calls, text messages and face-to-face invitation. In most countries, a reminder of participation was sent. The response rate, measured as the total respondents to the estimated eligible participants had important variations across countries going from 23% in China to 54% in South Africa [39].

## Participants

Adult community-dwelling residents with traumatic and non-traumatic SCI who provided informed consent and were able to respond the survey themselves were included in InSCI. Since building overall functioning scores was the core interest of this study, we used data from the 6'635 participants (out of 6'665 InSCI respondents from the selected European countries) who had at least one answer in the functioning items included in the analyses.

## Variables and tools

**Demographic and SCI characteristics.** Data on gender, age, age at the time of the injury, lesion level (paraplegia or tetraplegia), lesion completeness and time since SCI diagnosis (in years) are used.

**Functioning.** To construct an overall functioning score, the brief ICF Core Set for SCI in the long-term context was used as a reference framework [37]. We also took into consideration evidence highlighting that from a statistical perspective this Core Set should be augmented with additional ICF categories to better discriminate persons with SCI [40]. Items of the InSCI questionnaire operationalizing ICF categories of the functioning components (body functions, activities and participation) were selected by the authors and are presented in S2 Table. As the self-reported version of the SCI Independence Measure (SCIM) calculates the independence in performing activities of daily living, its items were dichotomized into 0 (no problem) and 1 (problem) to be in line with the items addressing intensity of problems [41]. The same dichotomization strategy was used for items of the Secondary Conditions Scale for SCI (SCS-SCI) [42].

**Environmental factors (EFs).** Fourteen items from the Nottwil Environmental Factors Inventory-Short Form (NEFI-SF) [43] addressing always the negative impact of climate, accessibility, attitudes and supports, products, financial issues, and services on the lives of

persons with SCI were used. Response options of items are: "not applicable," "no influence," "made my life a little harder," or "made my life a lot harder".

## Statistical analysis

Descriptive statistics were estimated for demographics and SCI characteristics in total and by country. For plotting functioning by age, age at the time of the injury, and time since injury, the categorization recommended by the International Spinal Cord Society was used [44]. All results are presented separately for countries with representative or convenience samples due to the high likelihood of bias in the latter.

**Development of overall functioning scores.** The multilevel structure of INSCI data (individuals grouped in countries) led to the choice of the hierarchical version of Generalized Partial Credit Model (GPCM). This is an application of the version proposed by Jong et al. for the Graded Response Model [45]. The GPCM is an Item Response Theory (IRT) model that relates each item to a common latent "ability" trait, in our case functioning (S1 File) [46]. For each person, the GPCM determines the person ability, while, for each item, discrimination and thresholds are estimated. Item's threshold indicates the point on the latent trait where a response in a category is more likely than the precedent category. Item discrimination is a measure of the differential capability of an item, with higher value favoring a higher response category more rapidly as the ability (latent trait) increases.

The hierarchical GPCM accommodates potential Differential Item Functioning (DIF) across countries. DIF occurs when two persons from different countries with the same functioning abilities might answer the InSCI questions differently [47]. The hierarchical (multi-group) structure of data is captured within the same GPCM model, by estimating the items thresholds for each country and linking them by setting the country item threshold as the overall mean threshold plus the country deviation. This also impose to hierarchical structure in the ability: the ability of a person from a specific country is given by country mean ability plus the person variance within his/her country. For the identification of latent variable (functioning), the sum of each country thresholds is set to 0 [45].

We cast the model in a Bayesian framework [48]. The additional file 2 shows the full specification of the prior distributions for each parameter. The model was implemented with Markov Chain Monte Carlo (MCMC) methods [49] using the software STAN 2.12 [50], while the pre and postprocessing of all analysis were done in R 4.0 [51].

For the convergence of the MCMC results, the Gelman and Rubin scale reduction factor of value 1 is expected for all parameters [49]. In addition, a formal model check of the hierarchical GPCM was carried out by comparing the posterior predicted totals at each MCMC iteration with the original data totals by means of posterior predictive 'p-values' (S2 File). Ideally, a well fitted model has p-values close to 0.5, so the observed total of response options in each item are neither under or over predicted by the fitted model [52].

Before applying the model, we evaluated the IRT assumptions, namely unidimensionality, local independency, and monotonicity. Unidimensionality was satisfied if the bifactor analysis on the polychoric correlation matrix shows that: a) all items load high on a general factor; and b) factor loadings of questions on the general factor exceed those of group factors determined based on permuted parallel analysis [53–55]. Items are local independent if residual correlation from a single factor confirmatory factor analysis is smaller than 0.25 [56]. If thresholds of any item were not ordered, monotonicity assumption was not satisfied and response options were collapsed as recommended.

The final functioning metric was transformed to range from 0 to 100, with 100 representing the best possible functioning level.

**Functioning trends.** For each country, a linear regression model was carried out to investigate whether the interval functioning score is associated with either chronological age, age at SCI or time since injury for persons with para- and persons with tetraplegia. No other variables were used for adjustment.

**Identification of environmental factors.** Using multiple linear regression, with functioning scores as dependent variable and EFs as regressors, while controlling for chronological age and time since injury, the relative importance of each EF was assessed based on Proportional Marginal Variance Decomposition technique (PMVD) (R package relaimpo) [57]. The relative importance of each EF specifies how much this variable contributes to the total R2 of the regression model, with total R2 denoting the variance in functioning scores explained by the EFs. The greater the relative importance, the more impact an EF has on the functioning scores. The PMVD technique overcomes the problem of correlation among EFs by the permutation of regressors when stepwise adding each EF in regression. Thus, the relative importance measures are concentrated on the regressors with high predictive power. Only respondents with no-missing data in functioning scores, chronological age, and lesion level were included, as an imputation of these variables was considered inappropriate. Missing information on EFs were imputed by the non-parameterized random forest method (R package missForest) [58]. Sensitivity analysis using only complete cases for all predictors was carried out to evaluate the robustness of the regression analysis.

## Results

Table 1 presents sample characteristics by country and in total. Across countries, participants were predominantly male, around 60% had a paraplegia and almost 80% had a traumatic SCI. Given the uneven distribution of gender and etiology, graphs displaying functioning patterns stratified by them had overlapping curves. Functioning patterns are therefore presented in the graphs stratified only by lesion level, where differences between the two strata were consistently observed. Country samples varied considerably in relation to mean chronological age (younger in Lithuania, Poland, and Romania) and time since injury. Variations must be understood considering that solely Norway, Germany, the Netherlands, Poland and Switzerland have representative samples.

### Development of overall functioning scores

The three GPCM assumptions were reasonably satisfied. Bifactor analyses showed a strong general factor pointing out to unidimensionality, but items assessing *Emotional functions*, *Energy and Drive*, *Bowel dysfunction*, *Bladder dysfunction*, and *Sexual function* loaded higher in their respective group factor than the general factor (S1 Fig). We nevertheless kept these items as they loaded as well high in the general factors. The Pearson correlation of the person's abilities produced by the GPCM without covariates with these items and GPCM without covariates and these items was of 0.98. The low percentage of residual correlations above 0.25 (1.90%) indicated that local independence assumption was satisfied. After collapsing response options of several items, all items satisfied the monotonicity assumption. Using one chain with 5000 iterations after burn-in, the MCMC convergence was supported by the Gelman and Rubin scale reduction factor between 0.99 and 1.001. The posterior predicted p-values ranged from 0.49 to 0.52 (very closed to 0.50) (S3 Table). Items' thresholds for all sample and items' discrimination are also reported in S4 Table.

### Ageing and functioning

For countries using convenience sampling, graphs must be interpreted in light of a high risk of bias. Also, the 95% CI strongly depends on sample size and is narrower and more precise for

**Table 1. Basic sociodemographic and lesion characteristics of European country study participants by collaborating country.**

| | | Gender | Chronolog-ical age | Education | Time since injury | Age at injury | Level of injury | | Completeness of injury | | Aetiology | |
|---|---|---|---|---|---|---|---|---|---|---|---|---|
| | Total N (%) | Female gender N (%) | Median age at time of survey (in years) (Q1, Q3) | Median years of education (Q1, Q3) | Median time since injury in years (Q1, Q3) | Median age at injury in years (Q1, Q3) | Paraplegia N (%) | Tetraplegia N (%) | Complete N (%) | Incomplete N (%) | Traumatic N (%) | Non-traumatic N (%) |
| Missing | - | 21 (0.3) | 17 (0.3) | 440 (6.6) | 199 (3) | 216 (3.3) | 176 (2.7) | 176 (2.7) | 278 (4.2) | 278 (4.2) | 92 (1.4) | 92 (1.4) |
| Total | 6635 | 1812 (27.4) | 53 (42, 64) | 13 (11, 16) | 11 (5, 21) | 35 (23, 52) | 3933 (60.9) | 2526 (39.1) | 2372 (37.3) | 3985 (62.7) | 5215 (79.7) | 1328 (20.3) |
| **Country** | | | | | | | | | | | | |
| Norway | 606 (9.1) | 192 (31.7) | 60 (45, 70) | 13 (9, 15) | 8 (4, 12) | 51 (36, 62) | 338 (58.6) | 239 (41.4) | 106 (18.1) | 481 (81.9) | 420 (69.7) | 183 (30.3) |
| Lithuania | 217 (3.3) | 81 (37.3) | 42 (35, 48) | 13 (12, 16) | 16 (7, 22) | 25 (20, 33) | 152 (70.4) | 64 (29.6) | 161 (74.5) | 55 (25.5) | 202 (93.5) | 14 (6.5) |
| Poland | 967 (14.6) | 164 (17) | 45 (37, 57) | 13 (11, 15) | 11 (6, 19) | 29 (22, 43) | 510 (53.5) | 444 (46.5) | 434 (45.3) | 525 (54.7) | 860 (89.2) | 104 (10.8) |
| Germany | 1617 (24.4) | 447 (28) | 56 (46, 65) | 13 (12, 16) | 9 (4, 17) | 42 (25, 56) | 780 (50.7) | 758 (49.3) | 524 (33.3) | 1051 (66.7) | 1234 (79.1) | 327 (20.9) |
| The Netherlands | 256 (3.9) | 86 (33.6) | 59 (50, 69) | 14 (11, 18) | 10 (4, 22) | 45 (28, 56) | 159 (62.1) | 97 (37.9) | 72 (28.3) | 182 (71.7) | 159 (62.6) | 95 (37.4) |
| France | 410 (6.2) | 111 (27.1) | 53 (41, 62) | 14 (11, 17) | 16 (6, 26) | 28 (20, 44) | 267 (66.3) | 136 (33.7) | 171 (42.4) | 232 (57.6) | 331 (81.3) | 76 (18.7) |
| Switzerland | 1527 (23) | 439 (28.7) | 58 (47, 68) | 13 (12, 16) | 16 (8, 28) | 34.5 (23, 52) | 1039 (69.5) | 456 (30.5) | 484 (36.1) | 857 (63.9) | 1198 (79.3) | 312 (20.7) |
| Italy | 203 (3.1) | 53 (26.1) | 51 (40, 60) | 13 (8, 13) | 10 (5, 17) | 36 (25, 51) | 149 (73.8) | 53 (26.2) | 76 (37.8) | 125 (62.2) | 141 (69.8) | 61 (30.2) |
| Spain | 417 (6.3) | 125 (30) | 51 (42, 61) | 12 (8, 17) | 14 (5, 24) | 31.5 (22, 47) | 255 (63) | 150 (37) | 183 (44.7) | 226 (55.3) | 320 (77.5) | 93 (22.5) |
| Greece | 199 (3) | 54 (27.1) | 46 (38, 56) | 12 (12, 16) | 13 (6, 22) | 28 (20, 40) | 135 (68.2) | 63 (31.8) | 91 (46.2) | 106 (53.8) | 170 (85.9) | 28 (14.1) |
| Romania | 216 (3.3) | 60 (27.8) | 37 (30, 46) | 12 (10, 14) | 5 (2, 13) | 28 (21, 38) | 149 (69.3) | 66 (30.7) | 70 (32.6) | 145 (67.4) | 180 (83.7) | 35 (16.3) |

the countries with large samples like Germany, while larger for countries with small samples, like Greece. As the overall functioning score has been generated with adjustments for country, it is directly comparable across countries, and follows the prior distribution, with a mean of 50 on a scale from 0 to 100. Overall mean functioning for countries are displayed separately for countries with representative (Fig 1A) or convenience (Fig 1B) samples. Norway has the highest and Germany the lowest functioning score among countries with representative samples. Among the countries using convenience samples, Lithuania has the highest and Italy the lowest functioning scores.

## Chronological age and functioning

Among countries with representative samples (Fig 2A), older chronological age is consistently associated with a decline in functioning for paraplegia but the extent and pattern of decline differs, being most pronounced in Germany and the Netherlands. Persons with tetraplegia have consistently lower levels of functioning than persons with paraplegia but an association to age is observed only in Germany and Norway. Among countries with convenience samples (Fig 2B), a similar trend is observed for paraplegia. Except for Lithuania and Spain, the older the person with tetraplegia, the lower the level of functioning. The most pronounced declines

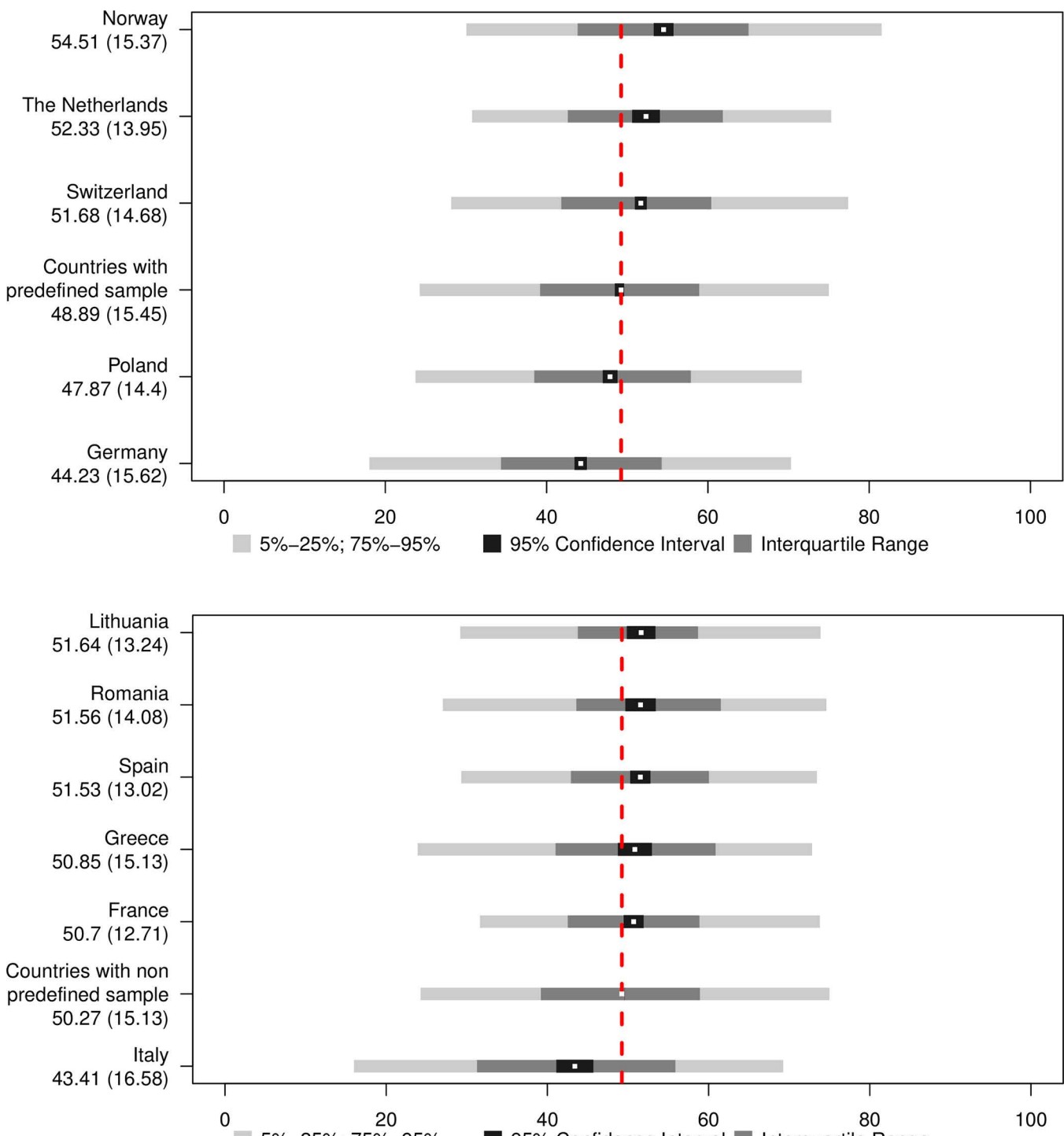

**Fig 1. a.** Distributions of overall functioning scores by country with a pre-defined sampling frame strategy and representative samples. The red dashed line cuts across the mean of the displayed countries. Mean and the standard deviation are displayed below the country name. The mean is also displayed in the graph by the white dots. **b.** Distributions of overall functioning scores by countries with convenience samples. The red dashed line cuts across the mean of the displayed countries Mean and the standard deviation are displayed below the country name. The mean is also displayed in the graph by the white dots.

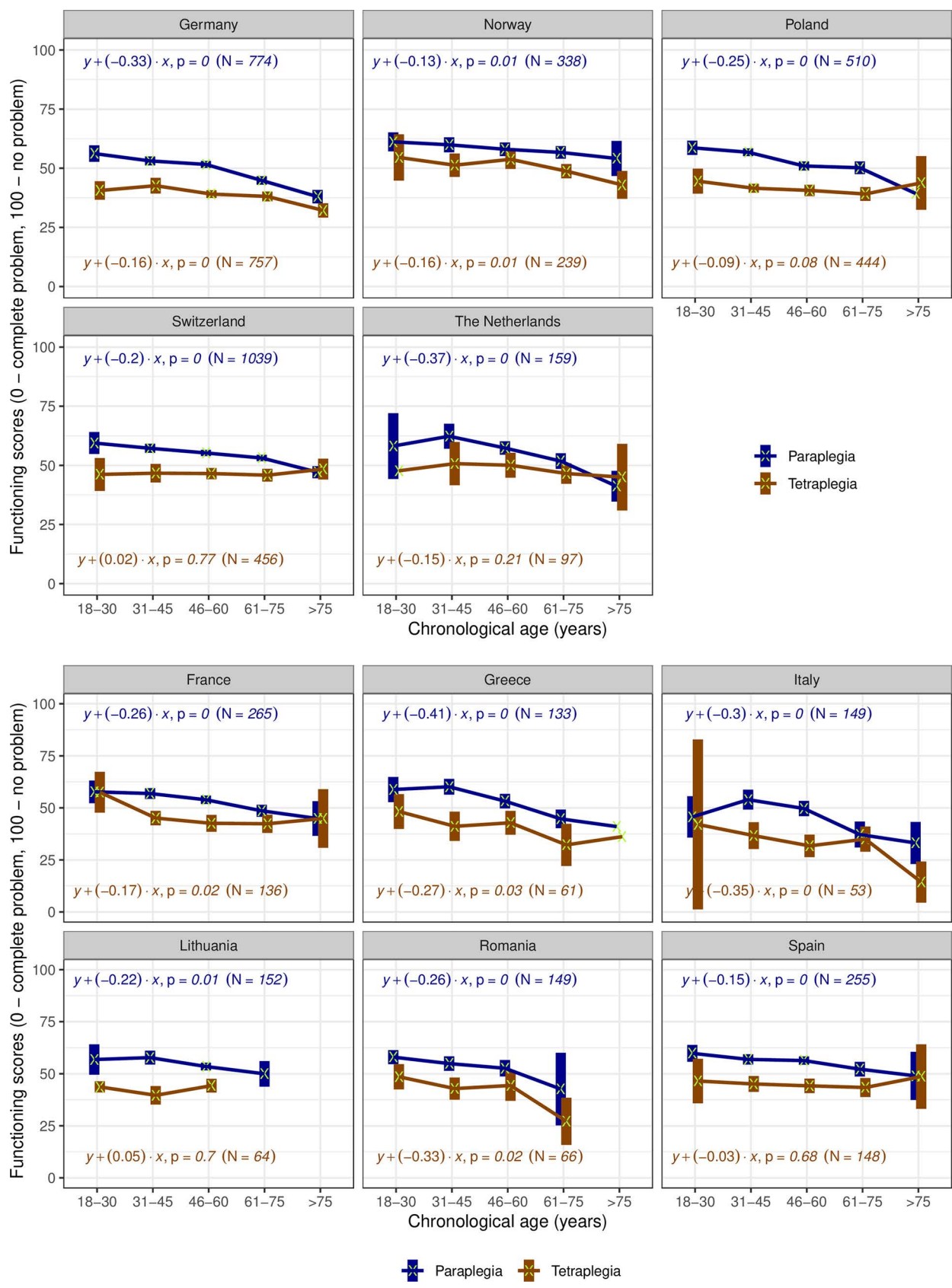

**Fig 2.   a.** Trends of functioning scores by chronological age groups and type of injury in countries with representative samples. For each group, the mean (marked with a green x) and its 95% Confidence Interval (the box around the mean) of functioning scores are displayed. The coefficient of the regression with functioning score as outcome (y) and continuous chronological age variable as predictor (x), their correspondent p-value and used number of cases are displayed for each country and lesion level (tetraplegia in blue versus paraplegia in brown). The groups where only mean is displayed have a sample of one person. **b.** Trends of functioning scores by chronological age groups and type of injury in countries with convenience samples. For each group, the mean (marked with a green x) and its 95% Confidence Interval (the box around the mean) of functioning scores are displayed. The coefficient of the regression with functioning score as outcome (y) and continuous chronological age variable as predictor (x), their correspondent p-value and used number of cases are displayed for each country and lesion level (tetraplegia in blue versus paraplegia in brown). The groups where only mean is displayed have a sample of 1 person.

in functioning are observed in Greece, Italy and Romania, for both lesion levels. Spain has a similar pattern to Poland and Switzerland.

## Age at injury and functioning

In countries with representative samples (Fig 3A) a frequent association between age at injury and current functioning level is observed. Although countries have different patterns, lower levels of functioning become more evident from a certain age at injury on. For instance, in Germany SCI sustained up to age 45 seems to have no association with the current level of functioning but from age 46 on the older the person was at injury, the lower the current functioning. Curves showing increased functioning level at very old age of injury, for instance in Poland, could reflect survival bias. In countries with convenience samples (Fig 3B), the association between age at injury and functioning is very consistent in paraplegia, with major differences across countries, for instance the very accentuated patterns in Italy and Romania. For tetraplegia, an association is observed only in Italy and Romania, and patterns across countries are very heterogeneous.

## Time since injury and functioning

An association between time since injury and functioning is not observed in most countries with representative samples (Fig 4A), neither for paraplegia nor for tetraplegia. When a trend is observed, declines are mostly small with the exception of persons living with tetraplegia in the Netherlands or in Switzerland for more than 15 and 25 years, respectively. In the countries with convenience samples (Fig 4B), no association is observed in France, Greece and Spain. Italy and Romania have counterintuitive patterns: the longer the time since injury, the better the functioning level, what points out again to a high probability of bias.

## Identification of environmental factors

For all countries with representative samples, problems with accessibility of homes of friends and relatives, access to public places and long-distance transportation were consistently among the top five determinants of functioning (Fig 5A), although the top predictor in Switzerland was climate. In the countries with convenience samples (Fig 5B), determinants are very country-specific. The most important barriers are nursing care and support services in Italy and Greece, communication devices in Romania, the climate in Lithuania, long-distance transportation in Spain and the accessibility of the homes of friends and relatives in France. The results of the sensitivity analysis corroborate these findings (S2 and S3 Figs).

## Discussion

This study used cross-sectional data collected in eleven European countries to describe functioning patterns of the SCI population in light of chronological age, age at the time of the

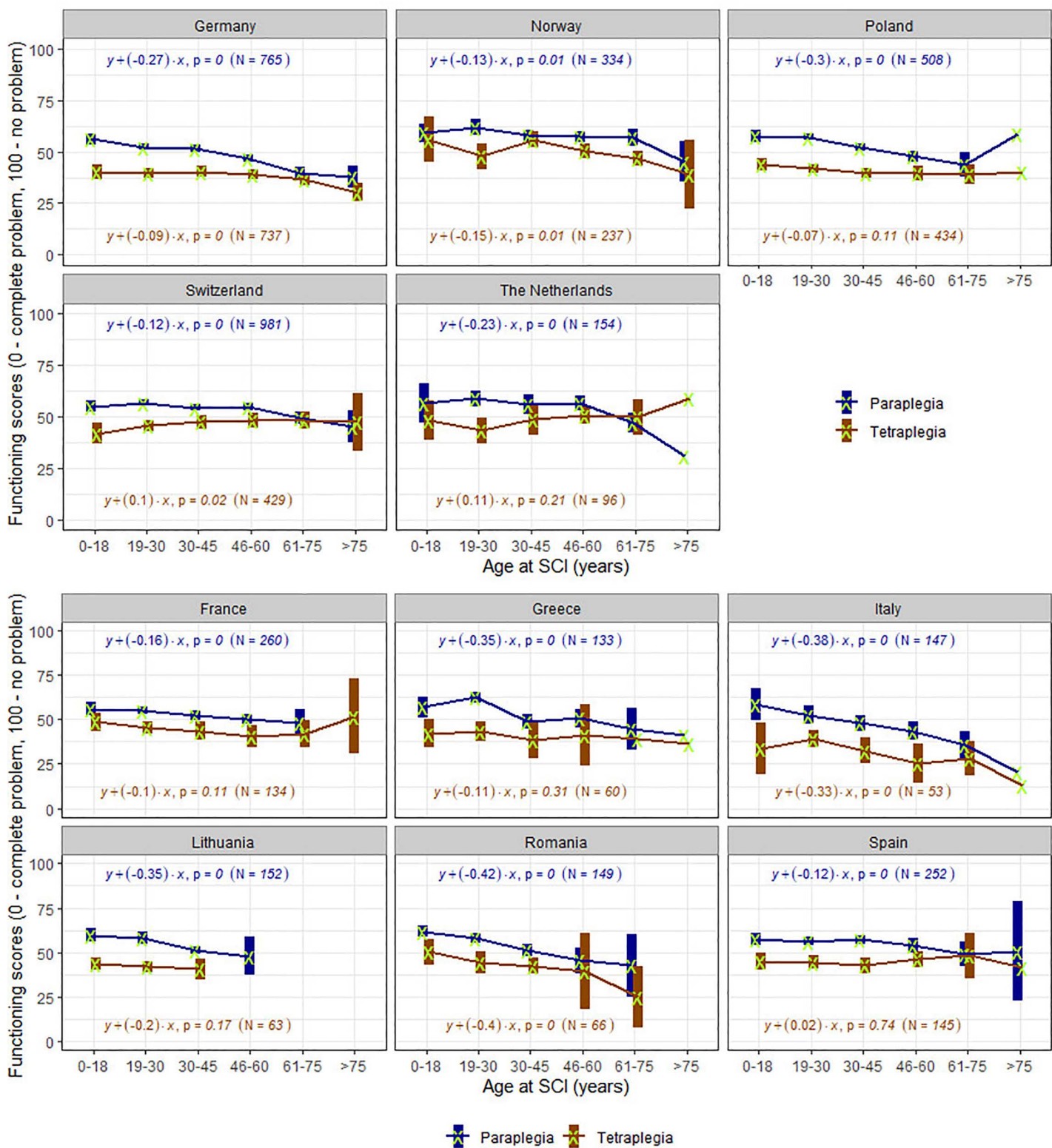

**Fig 3. a.** Trends of functioning scores by age at SCI groups and type of injury in countries with representative samples. For each group, the mean (marked with a green x) and its 95% Confidence Interval (the box around the mean) of functioning scores are displayed. The coefficient of the regression with functioning score as outcome (y) and continuous chronological age variable as predictor (x), their correspondent p-value and used number of cases are displayed for each country and lesion level (tetraplegia in blue versus paraplegia in brown). The groups where only mean is displayed have a sample of 1 person. **b.** Trends of functioning scores by

age at SCI groups and type of injury in countries with convenience samples. For each group, the mean (marked with a green x) and its 95% Confidence Interval (the box around the mean) of functioning scores are displayed. The coefficient of the regression with functioning score as outcome (y) and continuous chronological age variable as predictor (x), their correspondent p-value and used number of cases are displayed for each country and lesion level (tetraplegia in blue versus paraplegia in brown). The groups where only mean is displayed have a sample of 1 person.

injury, and time since injury as well as to identify environmental determinants of functioning. Enhancing methods traditionally used by WHO, e.g. [59] with machine learning techniques, we were able to create a common functioning metric with cardinal properties and to estimate corresponding overall scores of functioning comparable across the eleven countries. Driven most likely by the considerable resources and infrastructure needed to collect representative data, six countries used convenience samples and their results have been reported separately because of a high risk of bias.

Consistent with clinical experience, the main results of this study show that persons with tetraplegia, for all observed age groups, had a lower functioning than persons with paraplegia. In countries with representative samples–Norway, Germany, the Netherlands, Poland and Switzerland–older chronological age was consistently associated with a decline in functioning for paraplegia; for tetraplegia this association was only observed in Germany and Norway. Age at time of the injury and functioning level were associated, but patterns differed across countries. In general, lower levels of functioning became more evident from a certain age at injury on. An association between time since injury and functioning was not observed in most countries, neither for paraplegia nor for tetraplegia.

For the countries with convenience samples–Spain, France, Lithuania, Romania, Italy and Greece–the results show that functioning declined with chronological age for paraplegia. The most pronounced associations were observed in Greece, Italy and Romania, for both lesion levels. The association between age at injury and functioning was very consistent in paraplegia, with accentuated patterns in Italy and Romania. No association between functioning and time since the injury was observed in France, Greece and Spain. In contrast, in Italy and Romania the longer the time since injury, the better the functioning level, results that need to be carefully considered due to the high level of uncertainty in some of the results. This uncertainty is mostly driven by the small sample size in these two countries, which is more important when the results are disaggregated by age groups.

In terms of environmental factors, the lack of access to homes of friends and relatives, to public places, and to long-distance transportation were the most important determinants of functioning in Norway, Germany, the Netherlands, Poland and Switzerland. In contrast, in Italy and Greece, nursing care and support services were more relevant. In Romania, communication devices, in Lithuania was the climate, in Spain the lack of long-distance transportation, and in France the lack of accessibility to homes of friends and relatives.

Using functioning as a health indicator [60] suitable to condense the complexity of SCI, our results show that the association between chronological age and functioning declines over time. This result goes partly in line with related literature that describes the broad range of health losses, like reduced independence and an increase in secondary health conditions, that persons with SCI face as they age [17,19,61]. We consider it partly in line because in countries with representative samples, the association was consistently observed for paraplegia but not for tetraplegia. Nevertheless, the lack of association in the results for tetraplegia should be carefully considered for several reasons. First, the results for this group have more uncertainty; however, we consider that with more data the no association may go away and the patters in associations may resembled those observed for the paraplegic group. Second, the participants of InSCI are persons living in the community, we do not have data of persons with SCI newly

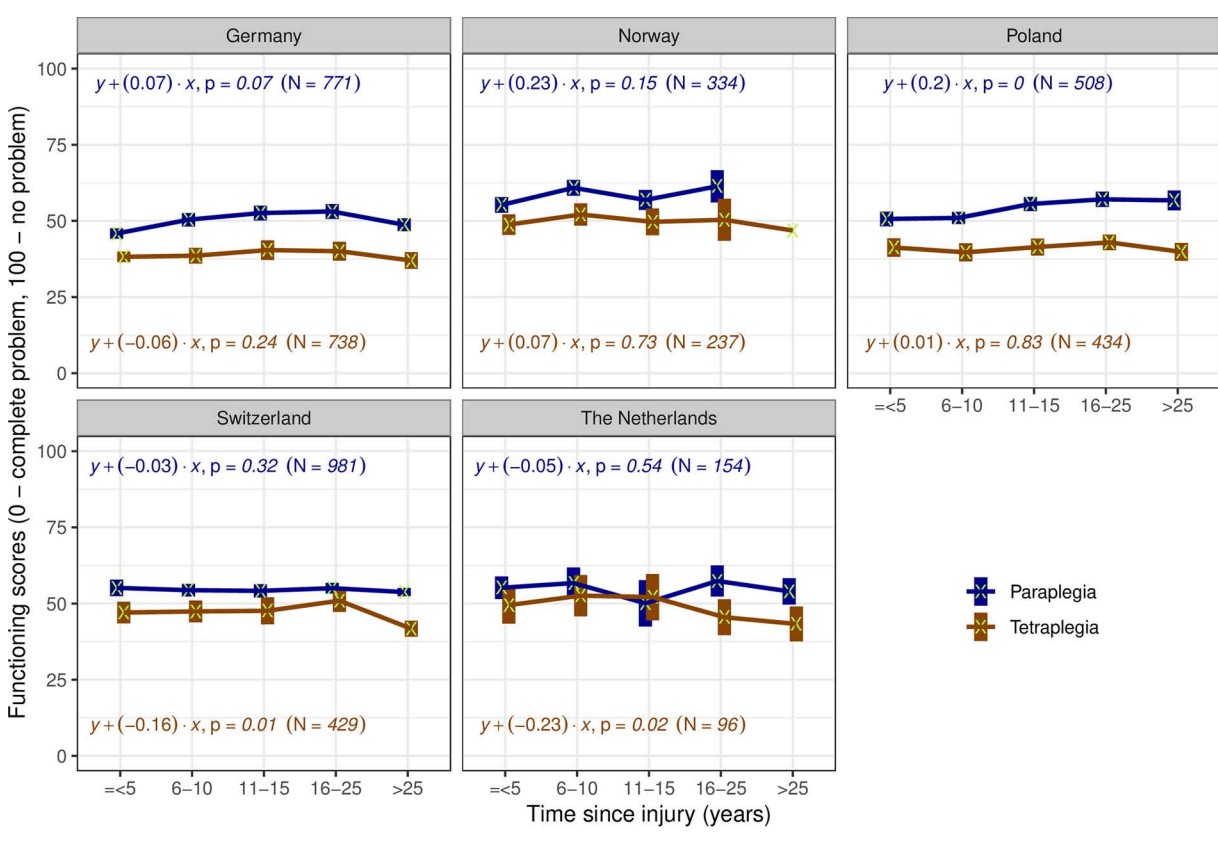

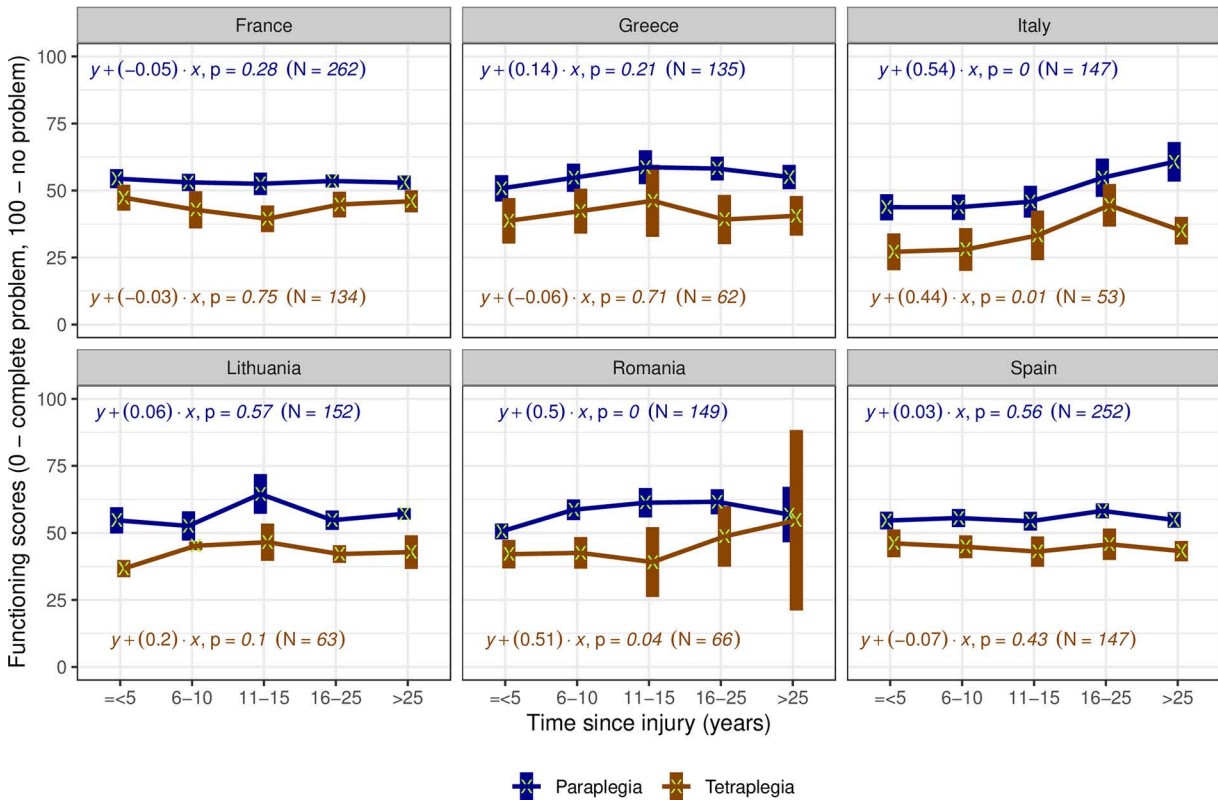

**Fig 4. a.** Trends of functioning scores by type since injury groups and type of injury in countries with representative samples. For each group, the mean (marked with a green x) and its 95% Confidence Interval (the box around the mean) of functioning scores are displayed. The coefficient of the regression with functioning score as outcome (y) and continuous chronological age variable as predictor (x), their correspondent p-value and used number of cases are displayed for each country and lesion level (tetraplegia in blue versus paraplegia in brown). The groups where only mean is displayed have a sample of 1 person. **b.** Trends of functioning scores by time since injury groups and type of injury in countries with convenience samples. For each group, the mean (marked with a green x) and its 95% Confidence Interval (the box around the mean) of functioning scores are displayed. The coefficient of the regression with functioning score as outcome (y) and continuous chronological age variable as predictor (x), their correspondent p-value and used number of cases are displayed for each country and lesion level (tetraplegia in blue versus paraplegia in brown).

injury. This implies our results describe the situation of those who have already gone through an improvement in their functioning levels. This is more important for the tetraplegic group, who have a more severe condition, and from whom the gains in functioning are potentially observed some months just after the injury. To expect significant gains over a long period of time may be less evident for this group. In fact, maintaining the functioning level may be an important gain for persons with tetraplegia. Finally, it is important to stressed that our results are cross-sectional, meaning that the results should be evaluated by group of age with no link over time. Hopefully with the second wave of InSCI, we can observe persons with SCI longitudinally to better understand how functioning and age are related.

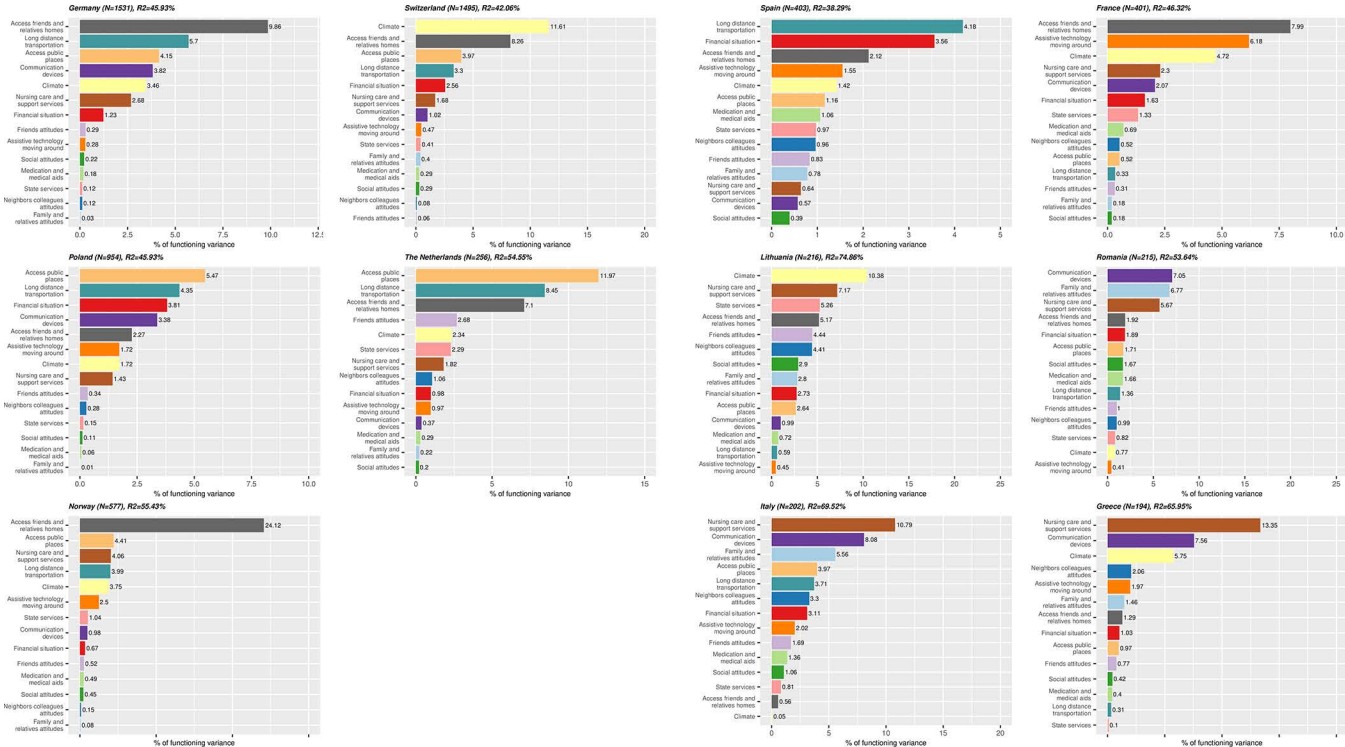

**Fig 5. a.** Relative importance of each environmental factors in explaining the total variation of functioning scores in countries with representative samples, when controlling for chronological age and type of injury. For each country, the full model variation when considering EFs and chronological age and type since injury as predictors is indicated in each country figure's title. **b.** Relative importance of each environmental factors in explaining the total variation of functioning scores countries with convenience samples, when controlling for chronological age and type of injury. For each country, the full model variation when considering EFs and chronological age and type since injury as predictors is indicated in each country figure's title.

Using a directly comparable functioning indicator and a common metric we have unveiled important differences across countries regarding the extent and the pattern of functioning declines by age. For instance, declines were more pronounced in Germany and the Netherlands than in Switzerland or Norway, raising the hypothesis on whether differences in rehabilitation organization and availability in the long-term context might be influencing the ageing process, a very important issue for clinical and policy stakeholders. The pronounced declines in functioning by age in Greece, Italy and Romania for both lesion levels are important but require, due to the convenience sampling, careful interpretation. Nevertheless, our results show that lack of or insufficient nursing care and support services have a high impact on functioning in those countries which might explain the observed declines in functioning.

In ageing research in SCI, age at injury and time since injury are considered highly relevant. Acquiring an SCI at a younger age has been linked to better recovery [10] for a myriad of reasons, such as the higher frequency of non-traumatic SCI and multimorbidity in older age. Our results of countries with representative samples confirm this assumption by showing that the older a person was at the time of the injury, the worse the current functioning level is, with different patterns across countries. Our results are also in line with evidence showing that lower age at onset was associated with better participation in individuals with SCI aged 65 years or older [62]. Time since injury is usually argued to be relevant because of exposition length to secondary health conditions, continuous intake of several medicaments, and reduced physical activity, among others reasons [13]. Nevertheless, our cross-sectional study failed to identify a consistent association between time since injury of persons with paraplegia or tetraplegia and functioning level in most countries with representative samples and identified contradictory patters in countries using convenience samples. Given the impact of the context of a person on functioning, an in-depth analysis exploring configurations of environmental determinants in different countries is recommended to better understand and explain patterns unveiled in the present study.

The built, social, attitudinal and political environment in which a person with SCI lives are key determinants of functioning. A previous study using InSCI data from 22 countries identified accessibility, climate, transportation, financial situation, and services provided by governments as the most important barriers [30]. Our results, using the functioning indicator as dependent variable, corroborates the major relevance of accessibility to homes of friends and relatives, access to public places, climate and long-distance transportation as cross-cutting barriers and complements the previous study with a direct ranking of impact on functioning by country. Our results are also in line with the evidence available for specific countries, for instance for Switzerland [31] or Germany [32].

The ranking of the impact of the environment on functioning is especially relevant for policy-makers to prioritize policies and public health actions. For example, in most of the countries in our sample, a starting point would be to promote or modify existing accessibility laws targeting infrastructure. House modifications or access to public spaces can have important benefits in the activity and participations of persons with SCI and other groups facing disability. To adapt private homes, financial support or inexpensive credit can be provided. Similarly, availability of adequate transportation is of extreme relevance as it is one of the main barriers to meet health care needs [63]. In many countries, regulations that require public transportation to be able to accommodate persons in a wheelchair already exist, but require a stricter enforcement. Finally, long-term care is a growing concern. Existing long-term care policies need to be adapted to cope with the increasing number of people in need of care. In the case of SCI, long-term care is even more relevant because requires some level of expertise, which is currently scarce and expensive. In some European countries, family members have the possibility to undergo training to provide adequate care [64]. Nevertheless, to identify what is the best way to approach

environmental factors is country specific. Thus, more research is needed, especially because environmental factors are key to achieving healthy ageing [65]. A starting point would be to integrate context factors in SCI ageing research agendas, from data collection to data analyses and reporting, using tools developed and validated for SCI like the NEFI [43].

In a Lancet series on ageing, Chatterji [26] used a functioning indicator to augment evidence on ageing patterns across countries obtained with morbidity and mortality, and strongly recommended the further study of the association between functioning patterns and contextual determinants. Current work on healthy ageing trajectories in the scope of the 2021–2030 the United Nations Decade of Healthy Ageing is completely based on the functioning indicator [27]. We followed these recommendations and trends, contributing to innovations in the SCI ageing agenda by introducing an indicator and a corresponding metric that can be used to model changes in health and functioning over time, as previously called for [18].

While the cross-sectional data we used is doubtless valuable for highlighting issues and informing health care planning and policy making, longitudinal data is fundamental for modelling functioning trajectories for persons with SCI as they age, monitoring trends and making causal inferences. A European SCI registry, similar to US National Spinal Cord Injury Database [66], would be helpful to overcome the challenges and problems faced by different countries to recruit representative samples and provide researchers and policy makers with the data needed for understanding and improving how persons with SCI live and age. In Europe, besides InSCI, there are other ongoing projects focusing on ageing with SCI that can also greatly benefit from such registry, especially when longitudinal data is envisioned [61].

This study needs to be understood in light of its limitations. First, six countries used convenience sampling. Reasons for having convenience samples include lack of infrastructure and of financial or human resources, among others, and for these countries this has been the "first time" ever that data for SCI was systematically collected at population level through an international standardized survey. Despite of the high risk of bias, this data provides preliminary insights and can motivate funders to support the recruitment of representative samples in the future. Second, due to sample size, we did not stratify the tetraplegia and paraplegia groups regarding lesion completeness. Nevertheless, we acknowledge that completeness of the injury might be an important factor that requires investigation in future research. Third, InSCI has a cross-sectional design and for a modelling of functioning over time as healthy ageing trajectories, longitudinal data is needed. Fourth, the analyses of contextual factors presented in this study is quite broad. A more in-depth analyses of environmental factors and their impact on functioning of persons with different SCI characteristics is strongly recommended as this is key to inform policy-making in the countries. Fifth, many more ICF categories are relevant to SCI, a complex condition, as the ones included in the brief Core Set for SCI in the long-term context that we used in this work, and clinical studies should use as many categories as possible to build functioning profiles. However, in epidemiological studies, we need to include as many items as needed to get a valid and precise score, but at the same time as less items as possible to ensure the feasibility of building a composite score using Item Response Theory. Finally, in the method used to develop a common metric for all countries the variances of the threshold parameters were set to not vary across items. This decision was motivated by the feasibility of the data analyses given the large sample and the very high computational time needed if the variances were set to vary.

## Conclusions

Functioning is a key health indicator, besides mortality and morbidity, and the fundament of ageing research. Enhancing methods traditionally used to develop metrics with Bayesian

approach, we were able to create a common metric of functioning with cardinal properties and to estimate overall scores comparable across countries. Focusing on functioning, our study complements epidemiological evidence on SCI-specific mortality and morbidity in Europe and identify initial targets for evidence-informed policy-making.

## Supporting information

**S1 Fig. Factor loading of functioning items on the general factor versus four factors identified by permuted parallel analysis.**
(PDF)

**S2 Fig. Relative importance of each environmental factors in explaining the total variation of functioning scores in countries with representative samples, when controlling for chronological age and type of injury.** Complete cases for all predictors were considered. For each country, the full model variation when considering EFs and chronological age and type since injury as predictors is indicated in each country figure's title.
(PDF)

**S3 Fig. Relative importance of each environmental factors in explaining the total variation of functioning scores countries with convenience samples, when controlling for chronological age and type of injury.** Complete cases for all predictors were considered. For each country, the full model variation when considering EFs and chronological age and type since injury as predictors is indicated in each country figure's title.
(PDF)

**S1 Table. Ethics committees or review boards approvals in the 11 International Spinal Cord Injury [InSCI] Community Survey countries.**
(DOCX)

**S2 Table. InSCI questions used in the development of functioning scores.**
(DOCX)

**S3 Table. Posterior predictive p-values corresponding to the observed totals.**
(DOCX)

**S4 Table. Items' discrimination and items thresholds for all sample.**
(DOCX)

**S1 File. The Hierarchical Generalized Partial Credit Model.**
(PDF)

**S2 File. Posterior predictive p-value.**
(PDF)

## Acknowledgments

This study is part of the InSCI Community Survey. InSCI provides the evidence for the Learning Health System for Spinal Cord Injury (LHS-SCI). See also Am J Phys Med Rehabil 2017;96(Suppl):S23-34). InSCI and the LHS-SCI are efforts to implement the recommendations of international perspectives on spinal cord injury (IPSCI) (Bickenbach JE, Officer A, Shakespeare T, von Groote P, editors. IPSCI. Geneva: WHO Press; 2013).

The members of the InSCI Steering Committee are Julia Patrick Engkasan (ISPRM representative), James Middleton (ISCoS representative, member scientific committee, Australia), Gerold Stucki (chair scientific committee), Mirjam Brach (representative coordinating

institute), Jerome Bickenbach (member scientific committee), Christine Fekete (member scientific committee), Christine Thyrian (representative study center), Linamara Battistella (Brazil), Jianan Li (China), Brigitte Perrouin-Verbe (France), Christoph Gutenbrunner (member scientific committee, Germany), Christina-Anastasia Rapidi (Greece), Luh Karunia Wahyuni (Indonesia), Mauro Zampolini (Italy), Eiichi Saitoh (Japan), Bum Suk Lee (Korea), Alvydas Juocevicius (Lithuania), Nazirah Hasnan (Malaysia), Abderrazak Hajjioui (Morocco), Marcel W.M. Post (member scientific committee, The Netherlands), Johan K. Stanghelle (Norway), Piotr Tederko (Poland), Daiana Popa (Romania), Conran Joseph (South Africa), Merce`Avellanet (Spain), Michael Baumberger (Switzerland), Apichana Kovindha (Thailand), and Reuben Escorpizo (Member Scientific Committee, United States).

## Author contributions

**Conceptualization:** Carla Sabariego, Cristina Ehrmann, Gerold Stucki.

**Formal analysis:** Cristina Ehrmann.

**Methodology:** Carla Sabariego, Cristina Ehrmann.

**Writing – original draft:** Carla Sabariego, Cristina Ehrmann, Gerold Stucki.

**Writing – review & editing:** Jerome Bickenbach, Diana Pacheco Barzallo, Annelie Schedin Leiulfsrud, Vegard Strøm, Rutger Osterthun, Piotr Tederko, Vanessa Seijas, Inge Eriks-Hoogland, Marc Le Fort, Miguel A. Gonzalez Viejo, Andrea Bökel, Daiana Popa, Yannis Dionyssiotis, Alessio Baricich, Alvydas Juocevicius, Paolo Amico.

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
