## [Decision Letter · Decision Letter 0]

14 Mar 2023

PONE-D-22-28841Ageing, functioning patterns and their environmental determinants in the Spinal Cord Injury (SCI) population: a comparative analysis across eleven European countries implementing the International Spinal Cord Injury Community Survey.PLOS ONE

Dear Dr. Sabariego,

Thank you for submitting your manuscript to PLOS ONE. After careful consideration, we feel that it has merit but does not fully meet PLOS ONE’s publication criteria as it currently stands. Therefore, we invite you to submit a revised version of the manuscript that addresses the points raised during the review process.

We look forward to receiving your revised manuscript.

Kind regards,

Giuseppe Barisano, M.D., Ph.D.

Guest Editor

PLOS ONE

Reviewers' comments:

Reviewer's Responses to Questions

**Comments to the Author**

1. Is the manuscript technically sound, and do the data support the conclusions?

Reviewer #1: Yes

2. Has the statistical analysis been performed appropriately and rigorously? 

Reviewer #1: Yes

3. Have the authors made all data underlying the findings in their manuscript fully available?

Reviewer #1: Yes

4. Is the manuscript presented in an intelligible fashion and written in standard English?

Reviewer #1: Yes

5. Review Comments to the Author

Attached.

---

## [Author Response · Author response to Decision Letter 0]

21 Mar 2023

Manuscript: Ageing, functioning patterns and their environmental determinants in the Spinal Cord Injury (SCI) population: a comparative analysis across eleven European countries implementing the International Spinal Cord Injury Community Survey

Preliminary notes

We would like to thank the Editor and the Reviewers for taking the time to review our paper. We received very insightful comments and suggestions, which have improved this version of the manuscript. We consider this has been a very enriching activity.

We have addressed each of comments and suggestions from the reviewers, and, within this letter, you will find our replies. To facilitate the revision, we have highlighted in yellow the changes done in the manuscript. This letter also indicates how the text in the manuscript changed.

We thank you for the opportunity to revise the previous version of the paper and hope this version will meet the publication standards of Plos One.

The authors

Reviewer 1:

Ageing, functioning patterns and their environmental determinants in the Spinal Cord Injury (SCI) population: a comparative analysis across eleven European countries implementing the International Spinal Cord Injury Community Survey.

Thank you for the opportunity to review this interesting paper! This study used data collected in eleven European countries to describe functioning patterns of the SCI population in light of chronological age, age at injury and time since injury as well as to identify environmental determinants of functioning.

Personally, I’m very impressed by the InSCI in general and this study doesn’t seem to be an exception. However, I have limited knowledge of the statistical methods used for developing the functioning metric and the PMWD technique, and this must be taken into account when reading this review.

The manuscript is well written and it seems as the work behind it is very thorough and rigorous. The authors provide a lot of background information to strengthen the methods they have used (however, I can’t seem to find additional file 2). They are transparent with limitations regarding representative data and it’s a strength that they report representative data and data from countries using convenience samples separately.

Specific comments:

Section: Introduction

1. Original line 93: ”a life” typo.

Response: change applied. See line 93.

2. Original line 97: “growth” typo.

Response: change applied. See line 97.

3. Original line 127: the authors could consider adding more information about why the aging process is important to study in the context of SCI, for example an increased vulnerability to age-related conditions and diseases, smaller margins to recover after an illness or injury etc.

Response:

Thank you for the comment. We agree with the reviewer. In general, the literature on ageing with SCI focuses on the relevance of differentiating between the health conditions related to the ageing process to the comorbidities related to SCI. This is very important to design adequate responses to persons with SCI in terms of prevention and treatment, especially now that we observe more older people with SCI. We have adapted the text in the manuscript as follows:

Lines 111-120:

An adequate response to the needs of persons ageing with SCI require comprehensive research on the topic (15). Ageing with SCI is complex because this group besides age-related conditions, have already a multi-morbid situation, both of which accelerate the ageing process (16). Currently, we observe that the mean age of the lesion is increasing due to an increment in the number of falls and domestic accidents. At the same time, the increasing life expectancy of persons with SCI rises the propensity of this group of facing more health complications (17). For these reasons, related literature emphasizes the importance to understand how the multiple complications observed on persons with SCI are related to their chronological age, to the age at the onset of the injury, and to the time with the injury (17, 18).

4. Original lines 152-155: As the authors write, aging with SCI is a complex process. I do not object to the fact that functioning is a key health indicator but the authors state that: “Given that the functioning indicator has the potential to capture the complexity of ageing with SCI in a single score, it is a highly relevant measure to understand the lived experience of ageing with SCI, …” The ICF conceptualizes a person's level of functioning as a dynamic interaction between her or his health conditions, environmental factors, and personal factors. In this study, the authors don’t touch upon the lack of personal factors in their functioning metric. Moreover, to “capture the complexity of ageing with SCI” and “understand the lived experience of ageing with SCI”, I believe a broader concept than functioning is needed. In my opinion there is also a need to take the persons’ subjective perspective and experiences into consideration. The ICF does not include QoL or more subjective measures of well-being, such as life satisfaction, which has been criticized. I would have liked to see a discussion about these issues (i.e., the exclusion of personal factors in the functioning metric and the lack of more subjective evaluations of the aging person’s life situation) if the authors indeed set out to use their metric to capture the complexity of aging and understand the lived experience of SCI. If not, please consider rephrasing this paragraph.

Response:

Thank you very much for raising these critical issues. We are using the ICF definition of functioning as the outcome of the interaction of health conditions and both personal and environmental factors. Following the WHO measurement tradition and the approach used in healthy ageing research, which are in line with the structure of the ICF with its two separate parts, one for functioning and one for contextual factors, we build a metric of functioning considering the components of the functioning part (body functions and structures, activities and participation), so that the metric can be then used to explore the impact of functioning determinants, i.e. personal and environmental factors, on the functioning of specific populations. It this sense, we not at all fail to appreciate the role of personal factors, which are an important determinant of functioning levels. In the present study, we examine environmental factors as predictors of functioning, controlling for chronological age and time since injury. In a planned second study, we would like to focus on personal-psychological factors as predictors of functioning, controlling for the impact of the environment using an overall score developed for the NEFI some years ago. The reason for a separation into two publications is the complexity of both contextual factors and personal factors. We fully agree with the reviewer that the current manuscript lacks a clear explanation of the approach we are using and have changed the text, as suggested, accordingly.

Line 148-151:

Additionally, functioning is understood as the outcome of the interaction between health conditions, personal-psychological factors (Geyh S, Müller R, Peter C, Bickenbach JE, Post MW, Stucki G, Cieza A. Capturing the psychologic-personal perspective in spinal cord injury. Am J Phys Med Rehabil. 2011 Nov;90(11 Suppl 2):S79-96.) and a range of features of the person’s context, such as the accessibility of the place of living, family support, social attitudes, and access to health care (24).

Lines 161-168:

The functioning indicator is built by condensing its components of body functions and structures, activities and participation domains, as defined in the ICF, into a single indicator and metric. This allows the study of the impact of its determinants, i.e. personal-psychological and environmental factors, on the level of functioning of specific populations. The functioning indicator has therefore the potential to contribute to the understanding of the complexity of ageing with SCI, enabling the identification of the its most relevant personal-psychological and environmental predictors.

Section: Methods

5. Original lines 204-205. Please provide more information about the InSCI, such as data collection processes, recruitment etc, maybe as a fact box or similar, or at least a reference.

Response:

Thank you for the comment. We have included more details about the data, and we also added the reference to the protocol paper of InSCI. The text has changed to the following:

Lines 209-220:

For the analysis, we used data collected in Norway, the Netherlands, France, Germany, Greece, Italy, Spain, Poland, Romania, Lithuania and Switzerland. As many countries lacked registries of persons with SCI, we only count with random samples from Norway, Germany, the Netherlands, Poland and Switzerland. For the remaining countries, convenient sampling was the only possibility to recruit participants, either through contact in health care facilities, government agencies and pre-existing databases from patient associations. To invite persons with SCI to participate in the survey, the national research teams used invitation letters, e-mails, phone calls, text messages and face-to-face invitation. In most countries, a reminder of participation was sent. The response rate, measured as the total respondents to the estimated eligible participants had important variations across countries going from 23% in China to 54% in South Africa (38).

38. Fekete C, Brach M, Ehrmann C, Post MW, Stucki G. Cohort profile of the International Spinal Cord Injury (InSCI) Community Survey implemented in 22 countries. Archives of Physical Medicine and Rehabilitation. 2020.

6. Original lines 217-218: I would have liked to see more information about how the authors reasoned when selecting items for their functioning metric:

- Was there a consensus discussion, did they include people with lived experience etc? For example, why not use the ISCoS Activity and Participation Data set? And there is nothing about respiration, infections, hospitalizations or sleep to mention a few other areas of relevance for people with SCI.

- There is also a need to include more information about the different assessment tools from which the authors extracted the items.

Response:

Thank you for the comments. We very much apologize that supplementary file 2 seems to have been forgotten, causing a lack of transparency about items selected, we have submitted it now. The table S2 reports all measures from which items were used. Regarding the selected InSCI items, the authors of the manuscript selected questions that operationalized the functioning components of body functions, activities and participation of the brief ICF Core Set for SCI in the long-term context, which was developed to reflect the most relevant ICF categories for this population after the acute phase. We complemented it with some additional categories suggested by Ballert C et al for epidemiological studies (Explanatory power does not equal clinical importance: study of the use of the Brief ICF Core Sets for Spinal Cord Injury with a purely statistical approach. Spinal Cord. 2012;50(10):734-9). Items and tools are shown in table S2. Respiratory function is not part of the brief ICF Core Set for SCI in the long-term context, we used questions about tiredness and feeling full of life to operationalize energy and drive, and neither infections nor hospitalizations belong to the functioning components (body functions, activities and participation), being rather highly important predictors of functioning. We fully agree that many more ICF categories are relevant to SCI, as the ones included in the brief Core Set for SCI in the long-term context, and that a clinical study should use as many categories as possible to build functioning profiles. However, for epidemiological studies, as ours, we need to include as many items as needed to get a valid and precise score (most relevant ones), but at the same time as less as possible to ensure the feasibility of building a composite score using Item Response Theory. The following sentence of the manuscript was accordingly revised and we have added the issue with the selection of items to the limitations of the study (lines 625-630):

Lines 235-238:

Items of the InSCI questionnaire operationalizing ICF categories of the functioning components (body functions, activities and participation) were selected by the authors and are presented in S2 Table.

Lines 625-630:

Fifth, many more ICF categories are relevant to SCI, a complex condition, as the ones included in the brief Core Set for SCI in the long-term context that we used in this work, and clinical studies should use as many categories as possible to build functioning profiles. However, in epidemiological studies, we need to include as many items as needed to get a valid and precise score, but at the same time as less items as possible to ensure the feasibility of building a composite score using Item Response Theory.

Section: Discussion

7. The authors acknowledge the lack of some expected findings, such as no association between age and lower levels of functioning in persons with tetraplegia and no association between time since injury and functioning. However, I would have liked to see a more elaborate discussion about possible underlying mechanisms for these findings, or lack thereof. Could it be related to the functioning metric, the study design or something else?

Response:

Thank you for the comment. We agree that these results are somehow unexpected, and need some discussion to better understand them. Thus, we have adapted the text to the following:

Lines 523-537:

Nevertheless, the lack of association in the results for tetraplegia should be carefully considered for several reasons. First, the results for this group have more uncertainty; we consider that with more data the no association may go away and the patterns in associations may resemble those observed for the paraplegic group. Second, the participants of InSCI are persons living in the community, we do not have data of persons with SCI newly injury. This implies that our results describe the situation of those who have already gone through an improvement in their functioning levels. This is more important for the tetraplegic group, who have a more severe condition, and for whom the gains in functioning are potentially observed some months just after the injury. To expect significant gains over a long period of time may be less evident for this group. In fact, maintaining the functioning level may be the most important gain for persons with tetraplegia. Finally, it is important to stressed that our results are cross-sectional, meaning that the results should be evaluated by group of age with no link over time. Hopefully with the second wave of InSCI, we can observe persons with SCI longitudinally to better understand how functioning and age are related.

8. Original line 540: How could these barriers be addressed by policy makers?

Response: We have now provided specific examples of how these barriers could be improved:

Lines 574-592:

The ranking of the impact of the environment on functioning is especially relevant for policy-makers to prioritize policies and public health actions. For example, in most of the countries in our sample, a starting point would be to promote or modify existing accessibility laws targeting infrastructure. House modifications or access to public spaces can have important benefits in the activity and participations of persons with SCI and other groups facing disability. To adapt private homes, financial support or inexpensive credit can be provided. Similarly, availability of adequate transportation is of extreme relevance as it is one of the main barriers to meet health care needs (62). In many countries, regulations that require public transportation to be able to accommodate persons in a wheelchair already exist, but require a stricter enforcement. Finally, long-term care is a growing concern. Existing long-term care policies need to be adapted to cope with the increasing number of people in need of care. In the case of SCI, long-term care is even more relevant because requires some level of expertise, which is currently scarce and expensive. In some European countries, family members have the possibility to undergo training to provide adequate care (63). Nevertheless, to identify what is the best way to approach environmental factors is country specific. Thus, more research is needed, especially because environmental factors are key to achieving healthy ageing (64). A starting point would be to integrate context factors in SCI ageing research agendas, from data collection to data analyses and reporting, using tools developed and validated for SCI like the NEFI (42).

9. Original lines 551-552: How, more specifically, do the results from this study contribute to innovations in the SCI aging agenda?

Response: In the cited reference, a very elaborated analyses of the state of art of ageing with SCI, one of the nine "recommendations to advance the field of aging with spinal cord injury" is to develop and use "new standardized measures to improve the assessment of changes in health and function and participation across the life span". The overall functioning indicator, which is very similar to the indicator used for modelling healthy ageing trajectories, has not, to the best of our knowledge, been used in SCI, despite of its potential. We see in the overall functioning indicator a measure that can indeed contribute to improving assessment and modelling of "changes in health and function and participation across the life span". To make this point clearer, we changed the mentioned sentence as follows.

Lines 598-600:

We followed these recommendations and trends, contributing to innovations in the SCI ageing agenda by introducing an indicator and a corresponding metric that can be used to model changes in health and functioning over time, as previously called for (18).

10. Original lines 552-560: There is an ongoing longitudinal project about aging with long-term SCI in Sweden with so far 12 publications, the authors could have referenced it: Jörgensen S, Iwarsson S, Norin L, Lexell J. The Swedish Aging with Spinal Cord Injury Study (SASCIS): Methodology and initial results. PM R. 2016;8(7):667-77.

Response: Thank you very much for the reference. Indeed, this is a very important project that need to be included in our literature. We have adapted the text as follows:

Lines 608-610:

In Europe, besides InSCI, there are other ongoing projects focusing on ageing with SCI that can also greatly benefit from such registry, especially when longitudinal data is envisioned (60).

11. Original line 497: From the abovementioned project, a recent study identifies declines in physical independence and an increase in certain secondary health conditions which could have been added to the discussion: Waller M, Jörgensen S, Lexell J. Changes over six years in secondary health conditions and activity limitations in older adults aging with long-term spinal cord injury. PM R 2022. doi: 10.1002/pmrj.12776

Response: Thank you very much for the reference. We have now included the main results of this project in the discussion.

Lines 514-516:

This result goes partly in line with related literature that describes the broad range of health losses, like reduced independence and an increase in secondary health conditions, that persons with SCI face as they age (17, 19, 60).

Section: Conclusion

12. Original lines 583-584: Referring back to my last comment regarding the introduction, “Having a focus how well persons with SCI live and age” seems a bit too broad for the functioning metric. I suggest the authors to rephrase in line with the wording in the conclusion in the abstract.

Response: Thank you for the comment, we agree and have followed the reviewer´s suggestion.

Lines 639-641:

Focusing on functioning, our study complements epidemiological evidence on SCI-specific mortality and morbidity in Europe, and identify initial targets for evidence-informed policy-making.

---

## [Editor Report · Decision Letter 1]

30 Mar 2023

Ageing, functioning patterns and their environmental determinants in the Spinal Cord Injury (SCI) population: a comparative analysis across eleven European countries implementing the International Spinal Cord Injury Community Survey.

PONE-D-22-28841R1

Dear Dr. Sabariego,

We’re pleased to inform you that your manuscript has been judged scientifically suitable for publication and will be formally accepted for publication once it meets all outstanding technical requirements.

Kind regards,

Giuseppe Barisano, M.D., Ph.D.

Guest Editor

PLOS ONE

---

## [Editor Report · Acceptance letter]

12 Apr 2023

PONE-D-22-28841R1

Ageing, functioning patterns and their environmental determinants in the spinal cord injury (SCI) population: a comparative analysis across eleven European countries implementing the International Spinal Cord Injury Community Survey

Dear Dr. Sabariego:

I'm pleased to inform you that your manuscript has been deemed suitable for publication in PLOS ONE. Congratulations! Your manuscript is now with our production department.

Kind regards,

on behalf of

Dr. Giuseppe Barisano

Guest Editor

PLOS ONE